


**Quantifying climatic influences on tree-ring width**
Guangqi Li[1], Sandy P. Harrison[1] and I. Colin Prentice[2,3,4]
[1]Department of Geography and Environmental Science, University of Reading, Reading, RG6 6AB,
UK
[2]AXA Chair of Biosphere and Climate Impacts, Department of Life Sciences, Imperial College
London, Silwood Park Campus, Buckhurst Road, Ascot SL5 7PY, UK
[3]Department of Biological Sciences, Macquarie University, North Ryde, NSW 2109, Australia
[4]Department of Earth System Science, Tsinghua University, Beijing 100084, China
**Abstract**
Before tree-ring series can be used to quantify climatic influences on growth, ontogenetic and
microenvironmental effects must be removed. Existing statistical detrending methods struggle to
eliminate bias, caused by the fact that older/larger trees are nearly always more abundantly sampled
during the most recent decades – which happens also to have seen the strongest environmental
changes. Here we develop a new approach to derive a productivity index ($P^*$) from tree-ring series.
The critical stem diameter, when an initial rapid increase in stem radial growth gives way to a gradual
decrease, is estimated using a theoretical approximation; previous growth rings are removed from
analysis. The subsequent dynamics of stem radial growth are assumed to be determined by: tree
diameter and height; $P^*$ (gross primary production per unit leaf area, discounted by a "tax" due to the
respiration and turnover of leaves and fine roots); and a quantity proportional to sapwood specific
respiration ($r_1$). The term $r_1$ depends not only on the growth rate but also on tree height, because a
given leaf area requires a greater volume of living sapwood to be maintained in taller trees. Height-
diameter relationships were estimated from independent observations. $P^*$ values were then estimated
from tree ring-width measurements on multiple trees, using a non-linear mixed-effects model in
which the random effect of individual tree identity accounts for the impact of local environmental
variability, due to soil or hydrological conditions, and canopy position (i.e. shading and competition).
Year-by-year $P^*$ at a site should then represent the influence of year-by-year changes in environment,
independently of the growth trend in individual trees. This approach was applied to tree-ring records
from two genera (*Picea* and *Pinus*) at 492 sites across the Northern Hemisphere extratropics. Using
a multiple linear mixed-effects regression with site as a random effect, it was found that estimated
annual $P^*$ values for both genera show consistent, temporally stable positive responses of $P^*$ to total
photosynthetically photon flux density during the growing season ($PPFD_5$) and soil moisture
availability (indexed by an estimate of the ratio of actual to potential evapotranspiration). The partial
effect of mean temperature during the growing season ($mGDD_5$) however was shown to follow a
unimodal curve, being positive in climates with $mGGD_5$ < 9 to 11 ˚C, and negative in warmer
climates.

**Keywords:** Tree-ring width, climate reconstruction, stem radial growth, ontogenetic trends,
productivity index, mixed-effects model, climate controls on tree growth



## 1 Introduction

The attribution of recent climate changes to anthropogenic influences (Hegerl et al., 1996; Hegerl and Zwiers, 2011; Bindoff et al., 2013) requires knowledge of climate changes before the instrumental period. Annually resolved palaeorecords of temperature change have been constructed for the past 2000 years (e.g. Mann el al., 2009; PAGES 2k Consortium, 2013; Anchukaitis et al., 2017). The most abundant source of information for such reconstructions comes from time series of tree-ring widths, because of their chronological accuracy, wide availability, and the sensitivity of tree growth to environmental factors. To use tree-ring data for palaeoclimate reconstruction, the raw data must be processed to remove ontogenic effects. In the absence of large environmental effects, radial growth shows an initial rapid increase but then passes a turning point and slows down gradually as the tree becomes larger and older. Removal of the ageing effect is called "detrending" in dendroclimatology. Various statistical methods have been developed for detrending (Cook, 1985; Fritts, 2012). Regional Curve Standardization (RCS; Briffa et al., 1996) has been widely recognized as a one of the best techniques for preserving trends due to longer-term environmental changes.

There remains however a fundamental, and worldwide, problem in the use of tree-ring data for climate reconstruction, namely the sampling bias of age distribution. The RCS method is not immune to this. Inevitably, field sampling focuses on long-lived trees in order to obtain as long a record as possible. Shorter-lived trees from earlier years have often disappeared by the time of sampling. Therefore there is a potential bias, such that growth rings from old, more slowly growing trees tend to be over-represented in the data for more recent years, while growth rings from younger, faster-growing trees tend to be over-represented in the data for earlier years. One potential consequence is that recent environmental trends in tree growth, which could be caused (for example) by the recent rapid increase in atmospheric $[CO_2]$ and/or global temperature increases, may be removed as an accidental by-product of detrending. The extent to which environmental effects are thereby missed in tree-ring reconstructions is controversial. For example, using the traditional method of constructing a ring width index (RWI), trees from the tropics apparently showed no effect of rising $[CO_2]$ since the 1950s despite consistently increasing water use efficiency, as shown by stable carbon isotopes (van der Sleen et al., 2015). However, Brienen et al. (2017) suggested that non-uniform recruitment effects on sampling could explain this "missing" $CO_2$ signal. They used a statistical correction accounting for biases in age at sampling, after which the $[CO_2]$ signal was significantly positive for both canopy and understory trees (Brienen et al., 2017).

There have been a number of suggestions of ways to deal with the problem of temporal bias, including using fossil trees in the detrending (e.g. Briffa et al., 1996) and (most obviously) more systematic sampling of both older and younger trees living today. Recognition of the potential bias (Briffa and Melvin, 2011) has led to several extensions of the RCS detrending method (e.g. cohort RCS: Esper et al., 2002; adaptive regional growth curves; Nicault et al., 2010; signal-free detrending: Melvin and Briffa, 2008). However, we argue that even if trees were sampled more systematically, there would still be sampling bias in the early record, because there is no way to resurrect old trees that are no longer available for sampling. Moreover, we require a reliable method to process the large volume of tree-ring data that are already available (https://www.ncdc.noaa.gov/data-access/paleoclimatology-data/datasets/tree-ring) as a source for climate reconstructions. For example, records covering the whole period of 1940-2000 for species of *Picea* and *Pinus* from the International Tree Ring Data Base are widely distributed across the northern extratropics (Fig. 1, left-hand panels). The right-hand panels of Fig. 1 show age distributions for some example sites. For 184 out of 188 *Picea* sites, the age of tree rings dated to the years 1970-2000 is greater than that for the years 1940-1969. There is a





difference of 26.4 years on average between the median ages of tree rings dated to the two periods.
For *Pinus*, 285 out of 304 sites showed the same phenomenon, with an average difference of 24.4
years (See Supplementary Information).

In this paper, we present a new approach to remove ontogenetic effects. Our approach makes use of
a limited amount of readily available independent information, and simple equations for the geometry
of tree growth, to infer values of an environmentally influenced productivity index ($P^*$). The use of
independent height and diameter data provides a useful biological constraint on the ontogenetic
growth trend. The inferred trend is related to size rather than age, reflecting the ergodic nature of tree
growth: that is, radial growth depends on tree size rather than directly on age, so – all else equal –
trees that start growing more rapidly also show a faster growth decline with age (Hättenschwiler et
al., 1997). In addition, we use the power of mixed-effects statistical modelling to separate random
effects on individual trees at a site from environmental (fixed) effects, which are presumed to
influence the growth of all of the trees at a site.

**2 Background**
**2.1 A simple diagnostic model for tree growth**
We derive a simple, generic model for tree growth in three steps as follows.
1. The annual increment of stem mass ($dW_s/dt$) is assumed to equal carbon export from the canopy
($X$), corrected for growth respiration, minus additional deductions. These are due to foliage turnover
and fine-root respiration and turnover, which are proportional to leaf area, and stem respiration, which
is proportional to sapwood volume and therefore – according to the pipe model (Shinozaki et al.,
1964) – to the product of leaf area and mean foliage height ($H_f$). $H_f$ is assumed to be $H/2$ (where $H$ is
tree height) initially, but must increase as the canopy rises. An expression for $H_f$ can be derived from
tree geometry as $H_f = H(1 - H/2aD)$, where $D$ is tree diameter, and $a$ is the initial slope of the height-
diameter relationship (Appendix A). Thus, we can write:

$$dW_s/dt = y\, A_c\, (X - qL)\, [1 - r_1\, H\, (1 - H/2aD)] \qquad (1)$$

where $y$ is the correction factor for growth respiration, $A_c$ is crown area, $X$ is carbon export from the
canopy, $q$ is a parameter related to foliage turnover and fine-root respiration and turnover, $L$ is the
leaf area index within the crown, and $r_1$ is a parameter related to the sapwood respiration rate. Note
the assumption here that the absolute respiration rate of sapwood varies in proportion to carbon
availability. This is inescapably true, at least to some approximation, as there is neither carbon
available nor a physiological need for suppressed trees to respire as much as dominant trees, or for
trees in poor growth years to respire as much as the same trees in good growth years.

2. The mass of a parabolic stem is related to its dimensions by $W_s = (\pi/8)\, \rho_s\, D^2 H$, where $\rho_s$ is wood
density. Hence the increase of stem mass with diameter is $dW_s/dD = (\pi/8)\, \rho_s\, D(2H + D dH/dD)$. If the
height-diameter relationship follows a Mitscherlich curve (Mitscherlich, 1928) then $dH/dD = a(1 -$
$H/H_m)$ where $H_m$ is the maximum height (Appendix B). We further assume $A_c = (\pi c/4a)\, DH$, where
$c$ is a constant related to the Huber value (i.e. the ratio of sapwood area to foliage area: Appendix A).
Therefore, $dW_s/dD = (\pi/8)\, \rho_s\, DH\, [2 + aD(1/H - 1/H_m)]$ and, applying the chain rule:
$$dW_s/dt = A_c\, (dD/dt)\, (a/c)\, \rho_s\, [1 + (aD/2)(1/H - 1/H_m)] \qquad (2)$$




3. Equating (1) and (2) and solving for d$D$/d$t$ yields:
d$D$/d$t$ = $P^* [1 - r_1 H (1 - H/2aD)]/ [1 + (aD/2)(1/H - 1/H_m)]$    (3)
where $P^* = (X - qL) (yc/a\rho_s)$.
The parameters $a$ and $H_m$ will be estimated from independent observations. Then $r_1$ and $P^*$ will be
estimated from all the tree-ring data for a given species and site. We will assume that $r_1$ depends only
on the species and site, while year-to-year variations in productivity for any given tree will be
reflected in $X$ and therefore also in $P^*$. We will also assume that there is a random effect on $P^*$,
corresponding to differences (genetic, or due to canopy position or microsite variation) among
individual trees.
An assumption implicit in the derivation of equation (2) above is that the trees have escaped the
initial, relatively short period of rapidly increasing diameter increment that is commonly observed.
Appendix C details the approximation we have used to estimate the ontogenetic turning point
corresponding to peak radial growth. Earlier rings are discarded.
**3 Data and methods**
**3.1. Tree-ring and climate data**
We use data from the two most widespread evergreen needleleaf tree genera in the northern
extratropics, *Picea* and *Pinus*, from the International Tree Ring Data Base (ITRDB,
https://www.ncdc.noaa.gov/data-access/paleoclimatology-data/datasets/tree-ring). All raw tree-ring
width data covering 1940-2000 CE were included in the analysis.
Gridded climate data for each site – 3-hourly air temperature (tas) and precipitation, downward short-
wave radiation (SWD), specific humidity and air pressure – were obtained from the WFDEI data at
0.5˚ resolution (Weedon et al., 2014). These data were further used for the calculation of annual total
photosynthetic photon flux density during the period with daily mean tas > 5°C ($PPFD_5$) and mean
growing-season temperature during the period with daily mean tas >5°C ($mGDD_5$). Monthly gridded
air temperature, precipitation, and cloud cover from CRU TS 3.23 (Harris et al., 2014) were used for
the calculation of the annual ratio of potential to actual evapotranspiration ($\alpha$) via the Simple Process-
Led Algorithms for Simulating Habitats (SPLASH) model (Davis et al. 2017). Note that $PPFD_5$
includes the effect of changes in growing season length as well as any change in average PPFD.
Annual values of all climate variables were calculated for the "effective carbon accumulation year",
conventionally defined as the period from 1 July in the year prior to ring formation to 30 June in the
year of ring formation.
**3.2 Estimating $a$ and $H_m$**
Values of $a$ and $H_m$ could in principle be estimated from local (site-level) field observations of the
diameters and heights of individual trees, across the full size range. However, it was not possible to
obtain local sets of paired observations of $D$ and $H$ for all the ITRDB sampling sites. Instead, we
estimated generic $a$ and $H_m$ ($H_{m(ICP)}$) values for *Picea* and *Pinus* using the available paired $D$ and $H$



measurements of each genus in the data set created by the Integrated Co-operative Programme on
Assessment and Monitoring of Air Pollution Effects on Forests (ICP Forests, http://icp-forests.net/).
We used 53,576 paired $D$ and $H$ measurements from 340 plots for *Pinus*, and 55,327 paired
measurements from 353 plots for *Picea* (Figure 2a, 2b). Maximum tree height tends to be fairly stable
within tree genera and species, but sites with especially low productivity (usually in extremely dry
and/or cold regions) may have deviating height-diameter curves, with low $H_m$. We therefore applied
a constraint for $H_m$ in such regions (Figure 2c). We performed a 99% quantile regression between the
satellite-derived observed maximum vegetation height (Simard et al., 2011) and the modelled long-
term mean GPP for 1982-2011 (Thomas, 2018), yielding an alternative estimate of maximum height,
$H_{m(sat)}$. For any given site, we assigned the lower of $H_{m(ICP)}$ and $H_{m(sat)}$ as the value of $H_m$. Only two
out of 269 *Picea* sites (< 1%) were affected by the $H_{m(sat)}$ constraint, but 83 out of the 151 *Pinus* sites
(55%) were affected. We tested the impact of uncertainties in the estimation of $H_m$ and $a$ on both
individual and site-level estimation of $P*$ by running sensitivity tests using a large (± 50%)
perturbation of both parameters.
**3.3 Estimation of $P*$**
We fitted a non-linear mixed-effects (NLME) model, based on equation (1), in two steps. The
response variable is the (post-peak) annual diameter increments (ring widths × 2) for all trees. In the
first step, the parameters to be estimated at each site are the fixed effect of $P*$ (site-level mean
productivity index during the whole period, a single value for a species and site), $r_1$ (also constant for
a species and site) and random effects corresponding to different $P*$ values for individual trees. The
contribution (weights) of trees' individual $P*$ values to the site mean $P*$ value, which accounts for
micro-environmental differences on the GPP of individual trees, and the site-level estimate of $r_1$, are
then used in the second step (a linear regression model) in order to estimate $P*$ for each year.
**3.4 Ring-width index calculation**
We also compared reconstructed $P*$ values with the standard calculation of the mean raw ring width
index (RWI-*mean*). Raw ring widths were first standardized as the ratio of ring width to the mean
ring width of each tree. The year-by-year values of RWI-*mean* at a site are then calculated as the
robust biweight mean of the standardised values of individual trees at that site (Cook and Kairiukstis
207 1990).

**3.5 Analysis of the bioclimatic controls on $P*$**
$P*$ is designed to reflect climate impacts on radial tree growth by removing the effects of sampling
biases, ontogeny and within-site variability. Individual bioclimatic parameters should therefore have
a consistent impact on growth, independent of sampling period or of genus considered. We used a
linear mixed-effects model to analyse the response of $P*$ to the bioclimate variables $mGDD_5$, $PPFD_5$,
and $\alpha$ for both *Pinus* and *Picea* during two intervals 1940-1969 and 1970-2000 CE. Site ID was
included as a random intercept. As the influence of $\alpha$ is strongly non-linear (Wang et al., 2017), $\alpha$
was transformed to natural logarithms before analysis.
During exploratory analysis, we discovered a strong interaction between the effect of $mGDD_5$
variability on $P*$ and the local mean value of $mGDD_5$. That is, the impact of warming on tree growth
was positive at low average temperatures, but negative at higher average temperatures. In subsequent





analyses we therefore hypothesized that the effect of $mGDD_5$ on $P*$ should follow a quadratic
(unimodal) curve, which can be expressed as:
$f(T) = \beta_0 + \beta_1\, mGDD_5 + \beta_2\, mGDD_5^2$
where $\beta_1$ is positive and $\beta_2$ negative. The turning point (maximum) of the unimodal response curve
occurs when $f(T) = -\beta_1/2\beta_2$. The uncertainty (standard error) of this derived value is estimated using
the variance-covariance matrix for $mGDD_5$ and $mGDD_5^2$ from the regression, as follows:

$$standard\ error\ of\ -\frac{\beta_2}{2\beta_1} = \sqrt{\frac{1}{4} \times \frac{\beta_1^2}{\beta_2^2} \times \left(\frac{Var(b)}{\beta_2^2} + \frac{Var(\beta_1)}{\beta_1^2} - 2 \times \frac{Cov(\beta_1, \beta_2)}{\beta_1 \times \beta_2}\right)}$$

We ran a further analysis to test whether the impact of recent changes in $[CO_2]$ were discernable in
the $P*$ reconstructions, by including $[CO_2]$ in addition to the bioclimatic variables ($mGDD_5$, $PPFD_5$,
and $\alpha$) in the linear mixed-effects model. We considered the two intervals 1940-1069 and 1970-2000
CE separately. The change in $[CO_2]$ over the first interval was *ca* 13 ppm, and over the second
interval *ca* 44 ppm.

**4 Results**
**4.1 Comparison of *P*\* and non-detrended ring widths**
The reconstructed time series of $P*$ at individual sites are comparable to raw ring widths over much
of the record. Differences between $P*$ and raw ring widths tend to be largest around the beginning
and end of each time series (Figure 3; Supplementary Information). Almost all $P*$ values are higher
than the RWI-*mean* (89% *Pinus*, 83% *Picea*) in the most recent 30 years, and lower (85% *Pinus*, 77%
*Picea*) in the earliest years. The difference between $P*$ and RWI-*mean* is most marked at sites which
show the largest sampling biases (e.g. CANA315, CO591, GERM189, TURK036) and least where
the sampling of large trees is not confined to the recent past (e.g. MOG039) or where the most recent
samples include some smaller trees (e.g. AK113). This comparison suggests that the calculation of
$P*$ has effectively reduced the effects of sampling biases as well as accounting for ontogeny.
**4.2 Sensitivity of inferred *P*\* to the imputed values of $H_m$ and *a***

Year-by-year $P*$ estimates are not highly sensitive to the selection of $H_m$ and *a*. Despite the large
range (50%) of $H_m$ and *a* values considered (Figure 4), their impact of the final year-by-year variation
of $P*$ was small. The correlation between alternative reconstructions is always > 0.98.
**4.3 Sensitivity of inferred *P*\* to the estimated peak-growth year**
Although a number of parameters are required to estimate the peak-growth year (Appendix C), most
of these are well constrained by observations (see Supplementary Information). The largest
uncertainties are associated with estimates of sapwood-specific respiration rate ($r_s$) and the ratio of
fine-root mass to foliage area ($\zeta$). Using a range of estimates for these parameters has a minor effect
on the identification of the peak-growth year, with minimum/maximum estimates indicating that peak
growth occurs when the diameter of the tree is between 2 and 11 cm for *Pinus* and between 5 and 12
cm for *Picea*. Differences caused by using our theoretical approximation, versus the simple





assumption that the first maximum in ring width corresponds to peak radial growth has little impact
on $P^*$ at most sites. The correlation between $P^*$ calculated using these two approaches to estimate
the peak at site TURK036, for example, is 0.97 (Figure 5; Supplementary Information). However,
the theoretical approximation makes it possible to calculate $P^*$ at sites where identification of peak
growth is problematic because the ontogenetic signal is conflated with that of environmental
variability.

**4.4 Inclusion of within-site variability on $P^*$**

The use of a mixed-effects model, including a random effect on productivity (random tree-to-tree
$P^*$), is the key to addressing the existence of cohorts of trees differing in productivity. The left-hand
panels of Figure 5 also show that the simulated growth trends differ among individuals. The
differences are shown for both the level of $P^*$ (initial productivity or growth rate), and the slope (a
combined effect of the growth rate and size-related tree geometry). These differences probably largely
reflect microenvironmental differences (including shading and soil depth variations) that are expected
to be more stable over time than climate effects during the life of an individual tree, except for
occasional gap-creation events. Therefore, the tree-to-tree random $P^*$ effects are carried over to the
second-step linear regression for each specific year's site-level $P^*$. This approach ensures that both
old and young trees have a similar influence on the final year-by-year time series of $P^*$ (in contrast
to conventional approaches that operate on the mean ring width).

**4.5 Global patterns of bioclimatic controls on $P^*$**

Consistent and significant patterns of bioclimate control on $P^*$ are shown for both genera, for
different periods, over the whole northern hemisphere (Table 1, Figure 6). Value of the slopes and
their ranges are stable within each genus among different periods (Table 1). Overall, $P^*$ shows a
significant positive response $\alpha$ (all $p$-values < 0.001), and a significant negative response to $mGDD_5$
(all $p$-values < 0.005). The response to $PPFD_5$ is also positive, and significant in three out of four of
the cases. The linear and quadratic terms of the response to temperature ($mGDD_5$) are both
consistently significant, with all $p$-values < 0.001. The consistency of the response through time, and
between genera, demonstrates that $P^*$ provides a robust estimate of year-to-year climate impacts on
radial growth and preserves the signal of long-term climate trends.
The consistently significant positive slopes for $mGDD_5$ and negative slopes for $mGDD_5^2$ demonstrate
the nonlinear impact of temperature on tree growth (Table 1, Figure 6). In cold climates, higher
temperatures in one year lead to increased radial growth but in temperate climates higher temperatures
have a detrimental effect on radial growth. The turning point is fairly consistent for genus and period,
being between 7 and 11°C for *Pinus* and 8 and 10°C for *Picea*. This consistency again supports the
idea that $P^*$ provides a robust estimate of the climate controls on tree radial growth. However, the
non-linear nature of this relationship challenges the conventional assumption of a monotonic
relationship that underpins most tree-ring based temperature reconstructions.
The impact of $[CO_2]$ on $P^*$ is equivocal (Figure 7, Table 2). There is a significant positive impact
during both intervals on *Picea*. There is positive (but not significant) impact on *Pinus* in the interval
1940-1969 CE but a significant negative trend during the interval 1970-2000 CE. All the effects are
very small compared to those of climate, and the bioclimatic relationships are stable compared to the
model without $[CO_2]$.



## 5 Discussion

The method described here removes ontogenetic effects from tree-ring records effectively and also accounts for microenvironmental differences (including e.g. shading and variability soil depth) on the productivity of individual trees. Its application is straightforward and requires only a modest amount of information external to the tree-ring records themselves. The key parameters, maximum tree height and the initial ratio of height to diameter, could be obtained for individual sites but can also be derived from regional forestry data. We have shown that the reconstructed $P^*$ is rather insensitive to uncertainties in the values used for these externally derived parameters, and to uncertainties in the estimation of the timing of peak radial growth.

However, $P^*$ does not solve the problem of making reconstructions of individual climate variables. The climatic control analysis indicates that several bioclimate variables simultaneously influence tree growth. Classically, reconstructions of past climates based on tree-ring series have focused on sites showing strong correlations with one particular climate variable. Our results suggest that this criterion can be, at best, only approximately fulfilled. Most importantly, even if the sensitivity of tree growth to one climate variable is strong in a certain range of that variable, that range is expected to be narrow. To take the case of temperature, the response is positive at lower growing-season mean temperatures but at higher temperatures it becomes flat, and then negative. The multifactorial (and potentially non-monotonic) nature of tree growth responses to climate is one of the proposed explanations for the "divergence problem" (D'Arrigo et al., 2008). This interpretation is supported by our results.

The $CO_2$ effect on tree growth remains enigmatic. A number of studies (Graumlich, 1991; Graybill and Idso, 1993; Gedalof and Berg, 2010; van der Sleen et al., 2015) have inferred that increasing [$CO_2$] has no impact on stem growth. Our method could, in principle, allow the detection of a $CO_2$ effect if present. We looked for such an effect (by including [$CO_2$] as an additional predictor in the climate response model), but no consistent response emerged. The apparent lack of a $CO_2$ effect in tree-ring records is puzzling, given that several Free Air Carbon dioxide Enrichment (FACE) experiments have shown $CO_2$-induced enhancement of tree growth (e.g. Oak Ridge FACE: Norby et al., 2002; DUKEFACE: McCarthy et al., 2010; Swiss FACE: Handa et al., 2006; EUROFACE: Calfapietra et al., 2003; Arizona FACE: Idso and Kimball, 1993). One possible explanation is confounding with other variables in the regression model (the fact that $CO_2$ has increased steadily means this is a real possibility when any other variable shows a unidirectional trend). This seems unlikely, however, given the stability of the bioclimatic relationships between the model with and without $CO_2$ included. Another explanation could be the counteracting effects of other environmental changes not considered, such as soil acidification. It is also possible that the $CO_2$ effect on stem growth is small because of increased carbon allocation below ground, as might be expected under nutrient-limited conditions in response to a $CO_2$-induced increase in nutrient demand. Increased below-ground allocation in response to increased $CO_2$ has indeed been observed both in laboratory experiments (Rogers et al., 1994; Prior et al., 2011) and in several of the FACE experiments (Oak Ridge FACE: Norby et al., 2004; DUKEFACE: DeLucia et al., 1999; Pritchard et al., 2008; Rhinelander ASPEN-FACE: King et al., 2001; EUROFACE: Calfapietra et al., 2003; Lukac et al., 2003; Bangor FACE: Smith et al., 2013).

The question of how best to use tree-ring data to reconstruct past climates remains open. One possibility would be to refrain from reconstructing single climate variables, and instead use $P^*$ as an index to compare with net primary production as simulated by ecosystem and Earth System Models for past climates. Another approach, echoing Fritts (2012), might involve the simultaneous use of

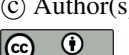


multiple $P^*$ reconstructions across a region to infer past changes in climate variability modes. The
detailed methods for such an analysis remain to be developed.

**Acknowledgements**
GL and SPH acknowledge support from the JPI-Belmont project "PAleao-Constraints on Monsoon
Evolution and Dynamics (PACMEDY)" through the UK Natural Environmental Research Council
(NERC) and from the ERC-funded project GC2.0 (Global Change 2.0: Unlocking the past for a
clearer future, grant number 694481). This research is a contribution to the AXA Chair Programme
in Biosphere and Climate Impacts and the Imperial College initiative on Grand Challenges in
Ecosystems and the Environment (ICP). We thank Dr. Tanja Sanders proving forestry measurements
from the ICP database and Dr. Rebecca Thomas for providing the $P$-model simulated long-term mean
GPP product.

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





### Appendix A

We assume that crown area ($A_c$) is related to diameter and height by $A_c = (\pi c/4a)\, DH$ where $D$ is diameter, $H$ is height, $a$ is the initial rate of increase of height with diameter, and $c$ is a constant. If $H$ follows a "diminishing return" relationship with $D$, as is always observed, this formulation ensures that the allometric coefficient of $A_c$ lies between 1 and 2. For small trees (with $H \approx aD$), which lack heartwood, the Huber value (ratio of sapwood area to foliage area) is then equal to $1/Lc$ where $L$ is the leaf area index within the crown (Li *et al.*, 2014). As the tree grows, however, height growth slows relative to diameter growth, and the tree's basal diameter becomes progressively greater than is needed to supply the leaves. For a tree of any size, we estimate the current height of the crown base ($z^*$) as the height ($z$) at which the ratio of stem area, $A_s\,(z)$ to foliage area, $A_f = L\, A_c = L\, (\pi c/4a)\, DH$ is equal to the Huber value. Using the unique property of paraboloid stems:

$$A_s\,(z) = A_s\,(0)\,(1 - z/H) = (\pi/4)\, D^2\,(1 - z/H) \tag{A1}$$

we obtain the solution:

$$z^* \;=\; H\,(1 - H/aD). \tag{A2}$$

We assume that the mean foliage height $H_f$ is the midpoint of $z^*$ and $H$:

$$H_f \;=\; H\,(1 - H/2aD). \tag{A3}$$

### Appendix B

The Mitscherlich curve (Mitscherlich, 1928) is a well-established "diminishing return" relationship, used in the forestry literature since the early 20[th] century. Applied to tree dimensions, its equation is:

$$H \;=\; H_m\,[1 - \exp\,(-aD/H_m)] \tag{B1}$$

where $H_m$ is the maximum height. This equation has the realistic properties (a) that the tangent at $D = 0$ is equal to the constant $a$, i.e. the relationship at first is approximately linear with slope $a$; (b) the rate of increase of $H$ with $D$ declines linearly with $H$, so that $dH/dD = a(1 - H/H_m)$; and (c) $H$ asymptotically approaches $H_m$ as $D \to \infty$. Allometric equations, in contrast, have none of these properties.

### Appendix C

The equation of the T tree-growth model (Li et al., 2014) is as follows:

$$dD/dt \;=\; \{y\,[P - \rho_s(1 - H/2aD)Hr_s/c] - L\,[1/\sigma\tau_f + \zeta(yr_r + 1/\tau_r)]\}/\{(a/2c)\,\rho_s\,[aD(1/H - 1/H_m) + 2] + (L/D)\,[aD(1/H - 1/H_m) + 1](1/\sigma + \zeta)\} \tag{C1}$$

The numerator is the total biomass production (per unit crown area), and the denominator is the biomass increment (per unit crown area) that is needed to generate a unit of diameter increment. $P$ is gross primary production (GPP, $kgC\ m^{-2}$), net of foliage respiration; $H$ (m) is tree height; $D$ (m) is stem diameter (with $H$ and $D$ linked by equation (2)); $y$ is the "yield factor" accounting for growth respiration; $L$ is the leaf area index within the crown; $\rho_s$ is sapwood density ($kgC\ m^{-3}$); $r_s$ is sapwood-




specific respiration rate (year$^{-1}$); $c$ is the initial ratio of crown area to stem cross-sectional area; $\sigma$ is
specific leaf area (m$^2$ kg$^{-1}$C); $\tau_f$ is foliage turnover time (years); $\zeta$ is ratio of fine-root mass to foliage
area (kgC m$^{-2}$); $r_r$ is the fine-root specific respiration rate (year$^{-1}$); and $\tau_r$ is the fine-root turnover time
(years).
To simplify equation (A1) we first note that the second term of the denominator quickly vanishes.
This term is responsible for the fact that peak diameter increment does not occur in year 1, but a few
years later. The division by $D$ however means that this term is important only near the start of growth
than it is later on.
The numerator describes two kinds of "tax" on GPP. The term including $L$ is a fixed rate (per unit
crown area) and corresponds to the allocation of GPP to foliage and fine-root turnover, and to root
respiration. The term including $r_s$ is initially small, but increases in proportion to height. These terms
affect tree growth differently. The term including $L$ represents a constant drain on GPP, while the
term including $r_s$ determines a size-related decline of biomass production.
The first term of the denominator is related to the wood density and the tree geometry. Because of
the flattening out of height growth, it initially takes the value $(3/2) (a/c) \rho_s$ but it declines somewhat,
eventually approaching the value $(a/c) \rho_s$.
If we disregard the second term of the denominator, it can be seen that root-related parameters ($\zeta$, $r_r$
and $\tau_r$) all appear together, thus the number of parameters to be estimated is reduced by two. We can
therefore write $Z = \zeta(yr_r + 1/\tau_r)$.
During early growth we can assume $H << H_m$, and therefore make the approximation $H \approx aD$.
Equation (A1) then simplifies to:
$dD/dt = \{y[P - (\rho_s r_s a/2c)D] - L_0\}/\{3a\rho_s/2c + 2Lm_A/D\}$          (C2)
where $L_0 = L[1/\sigma\tau_f + \zeta(yr_r + 1/\tau_r)]$, the total annual cost of making and maintaining leaves and fine
roots; and $m_A = (1/\sigma + \zeta)$, the leaf-plus-fine root mass per unit leaf area.
At the turning point, denoting the numerator of equation (A2) by $A$ and the denominator by $B$, we
have:
$B\, dA/dD = A\, dB/dD$          (C3)
where
$dA/dD = -y\rho_s r_s a/2c$          (C4)
and
$dB/dD = -(2L/D^2)m_A$          (C5)
By solving the resulting quadratic equation for $D$, it can be shown that the turning point occurs when:
$D = (4/3)(c/a\rho_s)Lm_A\{\sqrt{[1+(3/2)(yP - L_0)/(yr_sLm_A)]} - 1\}$          (C6)
This can be rewritten as:




$D = (4c/3a\rho_s)\, Lm_A[\sqrt{(1+P_x/u)}-1]$ (C7)
where $P_x = yP - L_0$ and $u = 2yr_s/3Lm_A$. Substituting this value into equation (C7) gives:
$dD/dt = (2c/3a\rho_s)\, \{P_x - u[\sqrt{(1+P_x/u)} - 1]/\{1 + 1/[\sqrt{(1 + P_x/u)} - 1]\}$ (C8)
Simplifying this further, define $Q = \sqrt{(1+P_x/u)} - 1$, hence
$P_x/u = (Q + 1)^2 - 1 = Q\,(Q + 2)$ (C9)
We can now derive the location of the turning point, which occurs when:
$D^2/(dD/dt) = 4cLm_A/(a\rho_s yr_s)$ (C10)
Thus, given an estimate of the maximum diameter growth rate, we can estimate the critical diameter
given suitable values of the parameters $c$, $a$, $\rho_s$, $L$, $m_A$, $y$ and $r_s$.
Note that the ratio of the maximum diameter growth rate to the area at which that growth rate is
achieved does *not* depend on the absolute value of the initial biomass growth rate ($P_x$). For high $P_x$,
this point will simply be reached sooner than for low $P_x$. However this ratio is greater for trees that
maintain greater initial crown area per unit basal area ($c$) and trees that maintain greater leaf and fine
root mass ($Lm_A$). It is less for trees that invest more in height growth (having high initial height-
diameter ratio $a$), have denser wood ($\rho_s$), and have intrinsically higher sapwood respiration rates ($r_s$).
Although these parameters are not known with precision, the change of the ratio of $D^2/(dD/dt)$ due to
variations in the parameter values is small. The quantity $D^2/(dD/dt)$ ranges from 0.097 to 2.59 at peak
radial growth of *Pinus*, corresponding to a diameter $D$ of between 2 to 11 cm when the maximum
stem growth $dD/dt$ is 5mm. The value of $D^2/(dD/dt)$ used in our theoretical approximation is 0.97.
Similarly, $D^2/(dD/dt)$ ranges from 0.49 to 3.11 at peak radial growth of *Picea*, corresponding to a
diameter $D$ of between 5 to 12 cm when the maximum stem growth $dD/dt$ is 5mm. The value of
$D^2/(dD/dt)$ used in our theoretical approximation is 1.31.





**Figure and Table Captions**

Figure 1. (a) Distribution of northern hemisphere *Pinus* and *Picea* tree-ring records covering period
1940-2000 (b) sampling age distribution through time for 3 *Picea* and 3 *Pinus* example sites, showing
age sampling biases. These six sites are identified on the map with letters (a-f).

Figure 2. Estimation of asymptotic maximum height ($H_m$) and the initial slope of height to diameter
(a). Panel (a) and (b) are the estimations for asymptotic maximum height ($H_m$) and the initial slope of
height to diameter (a) using all the measurements for *Picea* and *Pinus* from the Integrated Co-
operative Programme on Assessment and Monitoring of Air Pollution Effects on Forests  (ICP
Forests, http://icp-forests.net/). Panel (c) is 99% quantile quadratic regression using the satellite
observed maximum vegetation height (from Simard et al., 2011) and long-term mean gross primary
productivity (GPP) (from Thomas, 2018).

Figure 3. Comparison of $P^*$ and the standardised bi-weighted mean of the raw ring width (RWI-
*mean)* for the example sites in Figure 1.

Figure 4. Sensitivity of $P^*$ to maximum height *($H_m$)* and the initial slope of height to diameter *(a)*.
Panels a and b show the influence of different values of $H_m$ and a on the random effects of life-history
$P^*$ for each individual tree in the first step regression. Different coloured points are the measurements
for individual trees. Panels c and d show the comparison for the final site-level year-by-year $P^*$ with
different values of $H_m$ and *a*.

Figure 5. Effect of peak growth position on final year-by-year $P^*$: using the theoretical approach to
define peak radial growth (red) and using the first maximum ring width to define peak radial growth
(blue).

Figure 6. Observed response of $P^*$ to bioclimate variables. Partial residual plots, based on the linear
mixed model regression analysis, show the response of *Picea* and *Pinus* for the periods 1940-1969 and
617    1970-2000.


Figure 7. Observed response of $P^*$ to bioclimate variables and [$CO_2$]. Partial residual plots, based on
the linear mixed model regression analysis, show the response of *Picea* and *Pinus* for the periods 1940-
1969 and 1970-2000.

Table 1. Summary of the linear mixed model for the climate control analysis for the period of 1940-
1969, 1970-2000, and 1940-2000 for *Picea* and *Pinus*.

Table 2. Summary of the linear mixed model including $CO_2$ and climate for the period of 1940-1969,
1970-2000, and 1940-2000 for *Picea* and *Pinus*.



Table 1. Summary of the linear mixed model for the climate control analysis for the period of 1940-
1969, 1970-2000, and 1940-2000 for *Picea* and *Pinus*.

| species | period | Slope*10000 (except slope of PPFD$_5$*e+8) | | | | $R^2$_fixed | $R^2$_total | mGDD$_5$ at vertex (°C) |
| | | mGDD$_5$ | mGDD$_5^2$ | PPFD$_5$ | log($\alpha$) | | | |
| --- | --- | --- | --- | --- | --- | --- | --- | --- |
| *Pinus* | 1940-1969 | 2.65±1.32 | -0.17±0.05 | 10.43±3.48 | 24.76±1.45 | 0.059 | 0.882 | 7.59±1.6 |
| | 1970-2000 | 3.75±1.31 | -0.18±0.05 | 3.36±3.17 | 24.3±1.52 | 0.056 | 0.850 | 10.62±0.88 |
| | 1940-2000 | 3.18±1.03 | -0.18±0.04 | 1.32±2.63 | 23.04±1.14 | 0.071 | 0.846 | 8.61±0.99 |
| *Picea* | 1940-1969 | 5.01±1.71 | -0.27±0.08 | 11.6±4.32 | 23.09±1.65 | 0.067 | 0.880 | 9.18±0.74 |
| | 1970-2000 | 8.4±1.63 | -0.42±0.07 | 18.26±4.37 | 24.51±1.72 | 0.073 | 0.861 | 9.99±0.36 |
| | 1940-2000 | 6.81±1.24 | -0.36±0.05 | 7.62±3.27 | 22.14±1.24 | 0.070 | 0.859 | 9.43±0.38 |

Table 2. Summary of the linear mixed model including CO$_2$ and climate for the period of 1940-1969,
1970-2000, and 1940-2000 for *Picea* and *Pinus*.

| species | period | Slope*10000 (except slope of PPFD$_5$*e+8) | | | | | $R^2$_fixed | $R^2$_total | mGDD$_5$ at vertex (°C) |
| | | mGDD$_5$ | mGDD$_5^2$ | PPFD$_5$ | log($\alpha$) | CO$_2$ | | | |
| --- | --- | --- | --- | --- | --- | --- | --- | --- | --- |
| *Pinus* | 1940-1969 | 2.67±1.32 | -0.17±0.05 | 10.57±3.51 | 24.74±1.45 | 0.01±0.04 | 0.058 | 0.882 | 7.62±1.6 |
| | 1970-2000 | 3.63±1.31 | -0.17±0.05 | 5.94±3.28 | 25.16±1.55 | -0.03±0.01 | 0.052 | 0.847 | 10.77±0.89 |
| | 1940-2000 | 3.23±1.03 | -0.18±0.04 | 3.12±2.66 | 23.88±1.15 | -0.03±0.01 | 0.067 | 0.843 | 8.81±0.96 |
| *Picea* | 1940-1969 | 4.91±1.71 | -0.27±0.08 | 13.72±4.4 | 23.13±1.65 | 0.1±0.04 | 0.066 | 0.878 | 9.22±0.75 |
| | 1970-2000 | 8.29±1.63 | -0.42±0.07 | 13.76±4.57 | 23.68±1.74 | 0.04±0.01 | 0.073 | 0.865 | 9.82±0.38 |
| | 1940-2000 | 7.34±1.23 | -0.37±0.05 | 11.58±3.29 | 23.54±1.25 | -0.05±0.01 | 0.073 | 0.856 | 9.8±0.33 |




Figure 1. (a) Distribution of northern hemisphere *Pinus* and *Picea* tree-ring records covering period
1940-2000 (b) sampling age distribution through time for 3 *Picea* and 3 *Pinus* example sites, showing
age sampling biases. These six sites are identified on the map with letters (a-f).

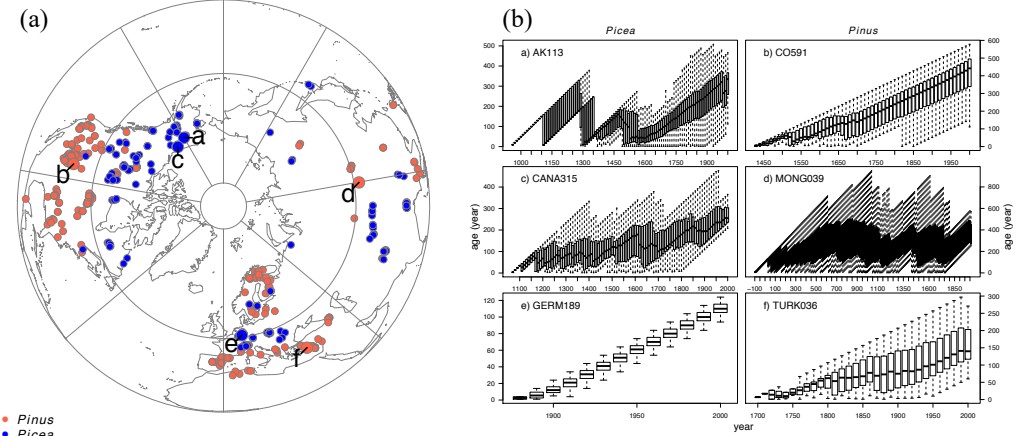



Figure 2. Estimation of asymptotic maximum height ($H_m$) and the initial slope of height to diameter
(a). Panel (a) and (b) are the estimations for asymptotic maximum height ($H_m$) and the initial slope of
height to diameter (a) using all the measurements for *Picea* and *Pinus* from the Integrated Co-
operative Programme on Assessment and Monitoring of Air Pollution Effects on Forests  (ICP
Forests, http://icp-forests.net/). Panel (c) is 99% quantile quadratic regression using the satellite
observed maximum vegetation height (from Simard et al., 2011) and long-term mean gross primary
productivity (GPP) (from Thomas, 2018).

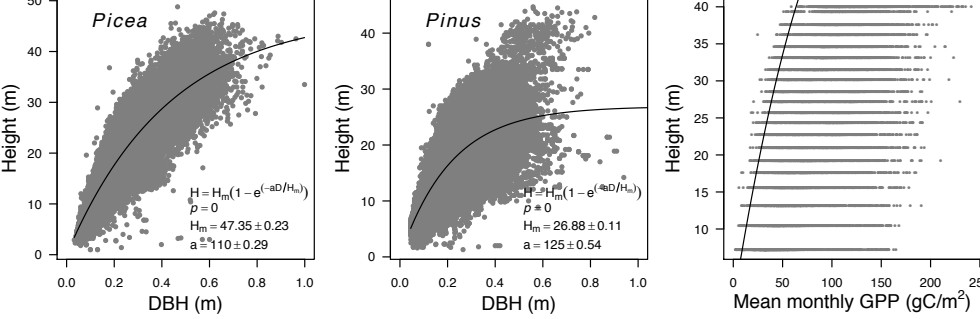







Figure 3. Comparison of *P\** and the standardised bi-weighted mean of the raw ring width (RWI-
*mean)* for the example sites in Figure 1.

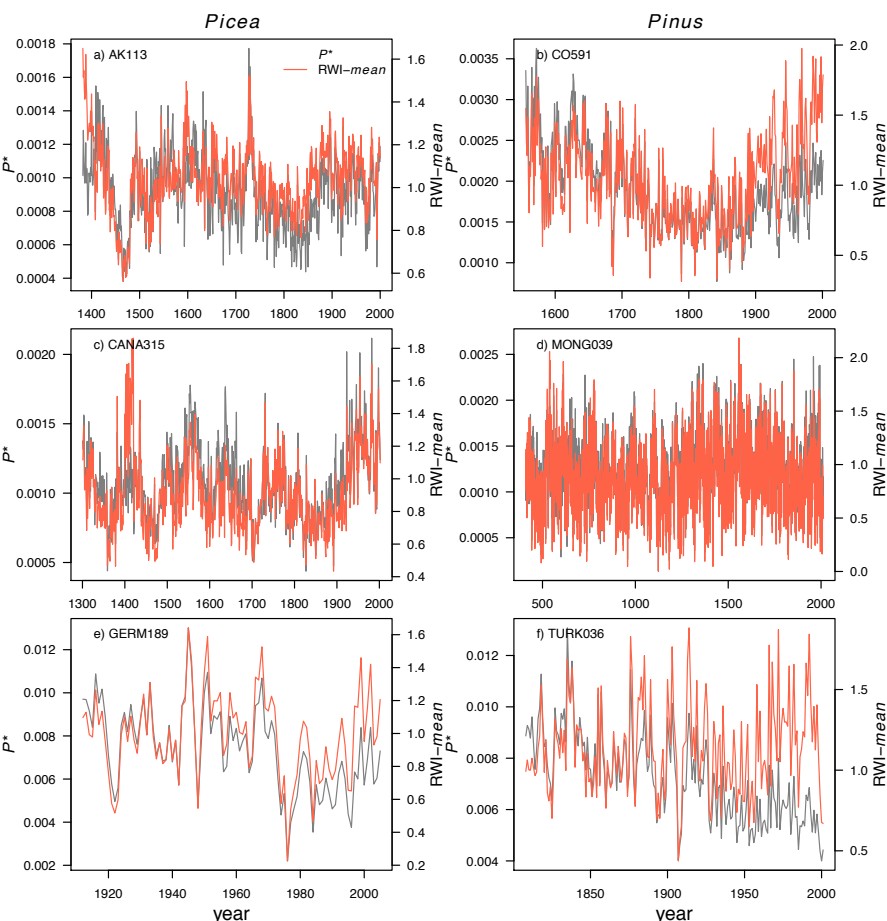






Figure 4. Sensitivity of $P^*$ to maximum height *($H_m$)* and the initial slope of height to diameter *(a)*.
Panels a and b show the influence of different values of $H_m$ and a on the random effects of life-history
$P^*$ for each individual tree in the first step regression. Different coloured points are the measurements
for individual trees. Panels c and d show the comparison for the final site-level year-by-year $P^*$ with
different values of $H_m$ and *a*.

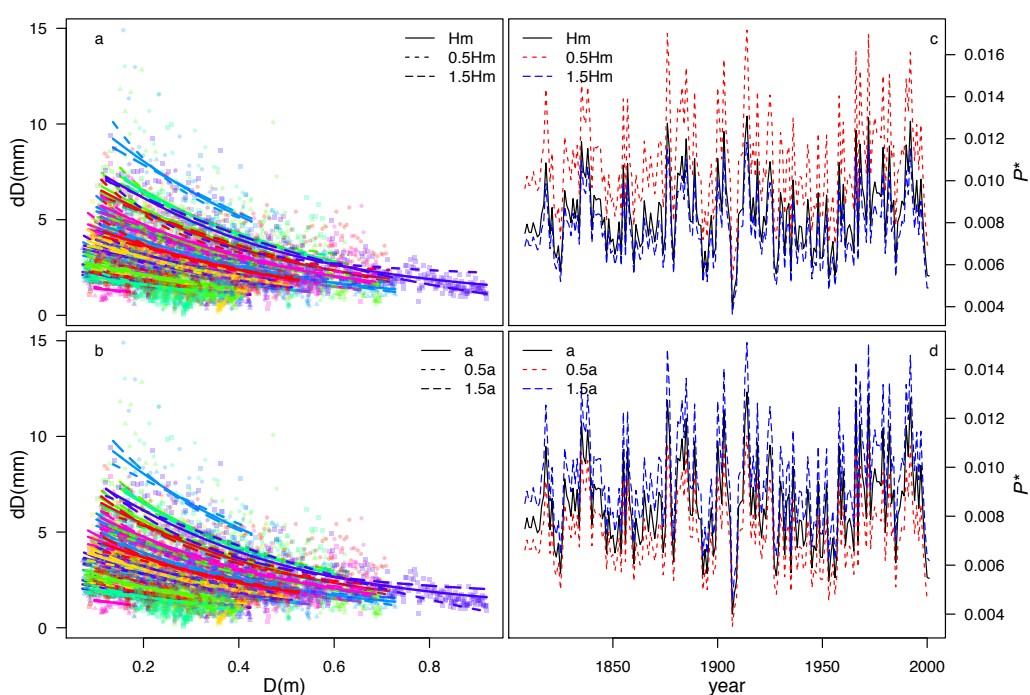






Figure 5. Effect of peak growth position on final year-by-year *P\*:* using the theoretical approach to
define peak radial growth (red) and using the first maximum ring width to define peak radial growth
(blue).

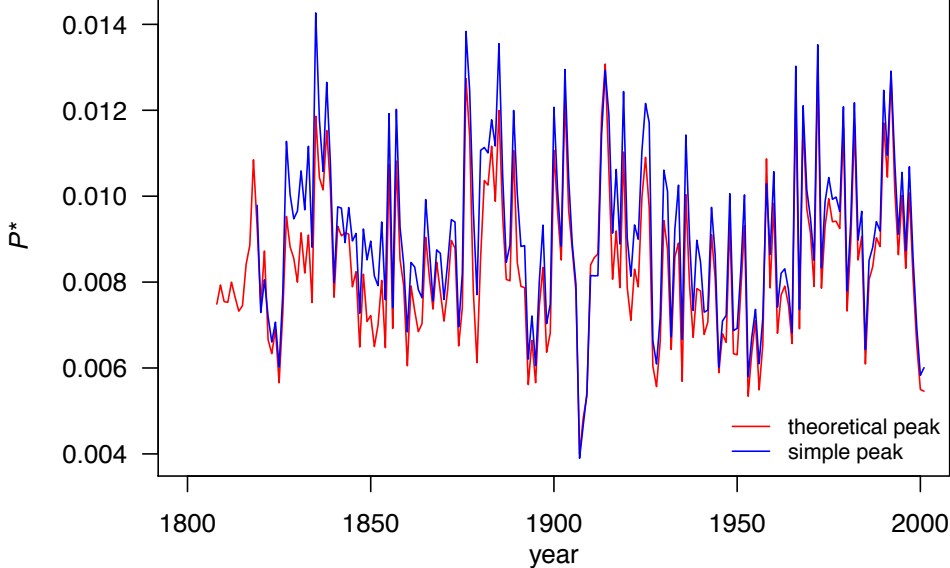






Figure 6. Observed response of *P\** to bioclimate variables. Partial residual plots, based on the linear
mixed model regression analysis, show the response of *Picea* and *Pinus* for the periods 1940-1969 and
675    1970-2000.

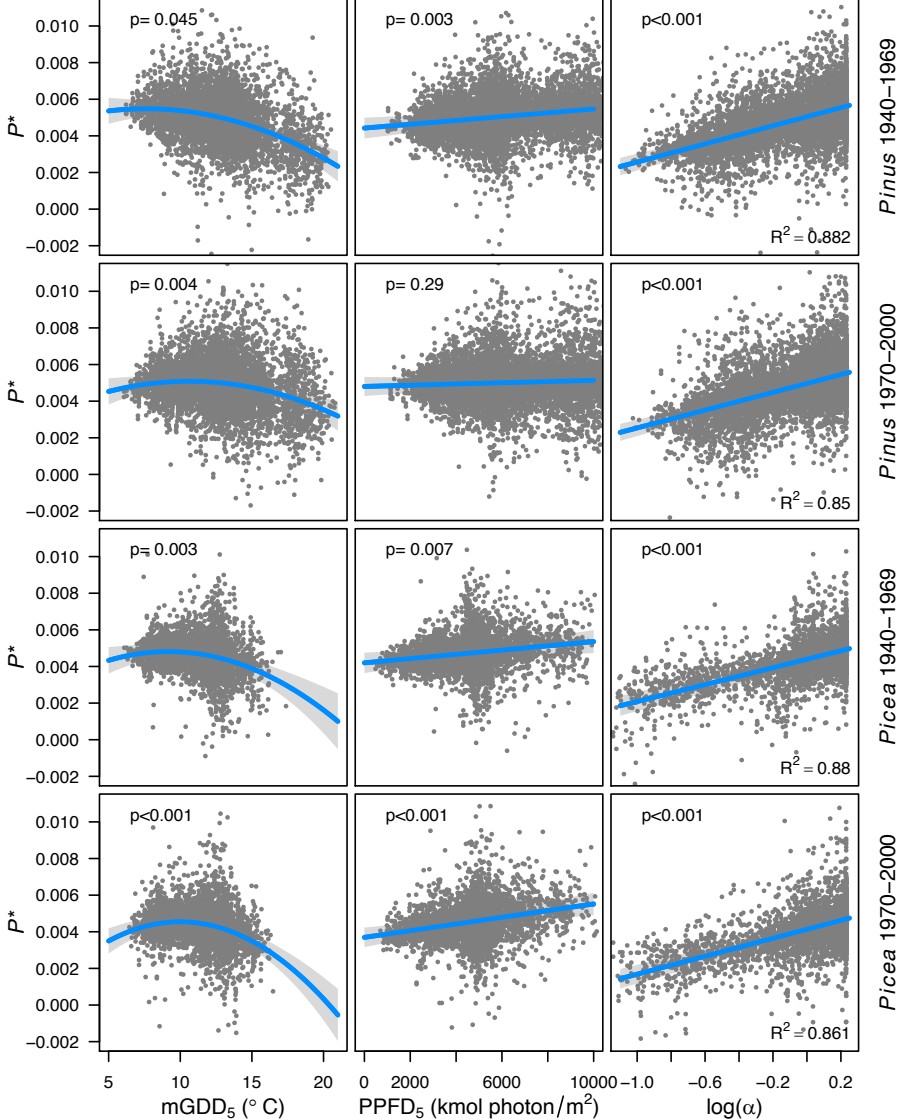






Figure 7. Observed response of $P^*$ to bioclimate variables and $[CO_2]$. Partial residual plots, based on
the linear mixed model regression analysis, show the response of *Picea* and *Pinus* for the periods 1940-
1969 and 1970-2000.

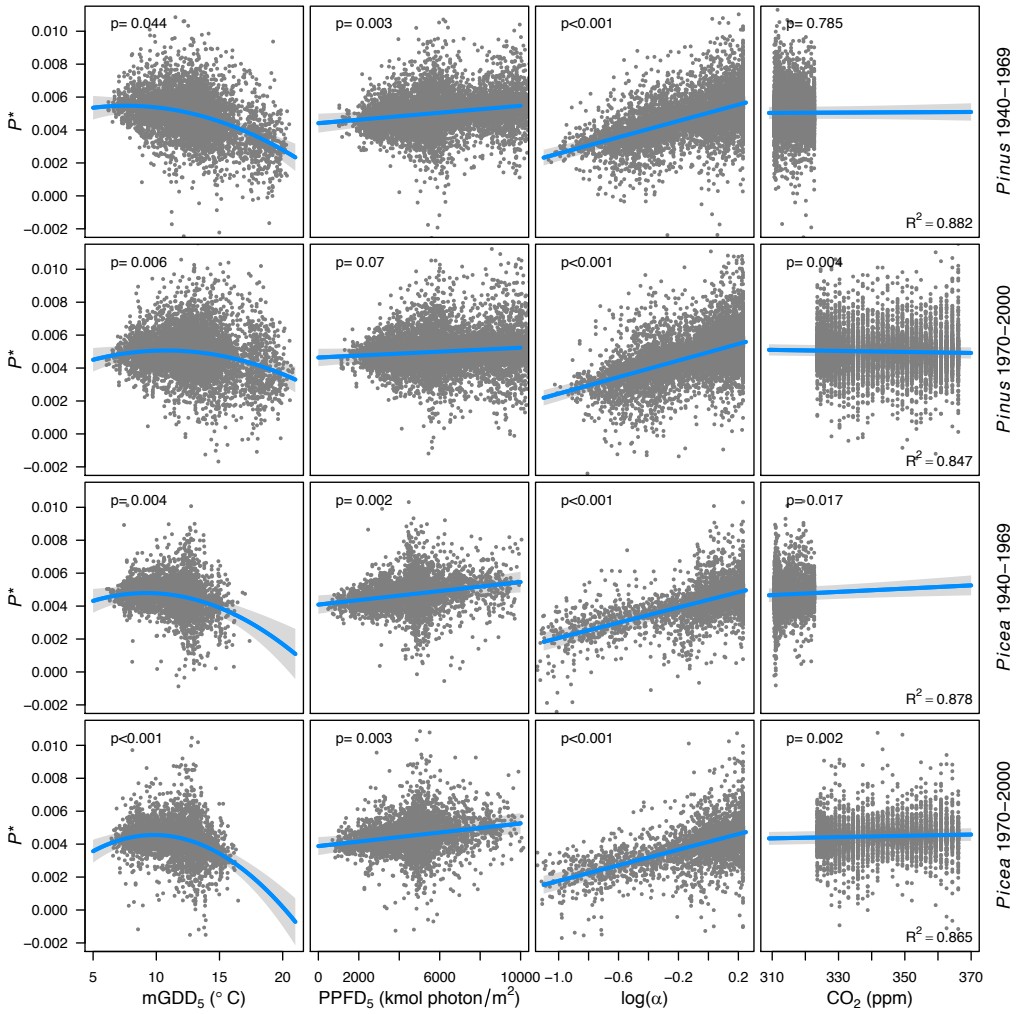
