# Peer review of "Discussion started: 13 March 2019 © Author(s) 2019. CC BY 4.0 License."

_Biogeosciences, 2019_

## Referee Comment (RC1) · Anonymous Referee #1 · 19 Apr 2019

Review bg-2019-63

The Li et al. paper presents a novel method to remove ontogenetic effects from tree-ring width series for the computation of a productivity index. Their advocate the need of a new method based on an existing sampling biais in classical detrending methodology caused by an over-representation of old trees in site-specific master chronologies at any given period. According to the authors this biais contaminate climate-growth correlations computed using detrended tree-ring series, especially those computed at site level. They provide a nice north-hemispheric analysis based on the *Picea* and *Pinus* tree-ring series extracted from the ITRDB.

GENERAL COMMENTS

This well written paper falls in the scope of the journal. I found pleasant to discover a new born methodology to handle tree-ring series in order to perform unbiased climate-growth analyses. However, even if this novel method looks appealing from a first perspective, it requires quite a substantial amount of parameters to be applied, compared to conventional methods (L307). I am not convinced that the application of this novel method is straightforward, despite what the authors say (cf. L307), particularly for other less common tree species or genii.

The high positive correlation between P* and RWI makes me think again that the classical detrending applied in dendrochronology would still be easier and simpler to use. To really advocate in favor of the use of the herein presented method, it would require to compare the climate-growth analyses presented in Figs 6 and 7 with similar climate-growth analyses computed with 'traditionally detrended RWI.

I have some major concerns that I hereafter discuss:

- Study area and environmental gradient
I am skeptical regarding the gradient over which they perform their analysis. The authors take two conifer tree genus from the northern hemisphere as examples, *Pinus* and *Picea* species. The tree populations include sites and pine and spruce species from the mediterranean, temperate and boreal contexts that respond quite differently to climate (e.g. *Pinus laricio* spp. vs *Pinus sylvestris*). For example, it is well known that growth responses to summer temperature move from being negative in warm and dry contexts (mediterranean biome) to positive in cold and wet environment (boreal biome). So I would like to see the analyses, especially the climate-growth analyses, ran for each genus over the three above-mentionned biomes.
This would be even more needed as the authors assume that "bioclimatic parameters have a consistent impact on growth" (cf. L212-213). This assumption is wrong, and it is even more wrong to assume that this is true regardless of sampling period or targeted tree species (cf. L213). Growth responses to climate are by nature not constant through time and space for a given tree species, due to non-linear interactions between environmental factors (e.g. climate and soil). The authors mention themselves (cf. L 220-221) that $mGDD_5$ has a non-linear impact on P*.

- 'the sampling biais of age distribution'
The authors ground their study in correcting the under-representation of young trees in growth chronologies. Therefore I have hard times to understand why the authors remove all rings before the turning point in growth trends (i.e. all juvenile rings). By doing so, they also contribute to the sampling biais of age distribution in their site chronologies. To me it seems that this removal of juvenile growth rings is only done to allow the approximation done in Appendix C. In traditional detrending, juvenile growth rings are kept.
A question arises: What is the average number of years (juvenile rings) removed by this step? How does age distribution look like after this step compare to the original age distribution (Fig. 1b)?

- Simplification of equations
The authors are simplifying quite substantially the equations they present in Appendix C based on different assumptions. Would these assumptions be applicable to other tree species, e.g. broadleaves species? If not, the herein presented method couldn't be applied to some tree species, and therefore would be useless in an more isolated geographical context.

- Detrending of tree-ring series

The authors use a mean fonction to detrend raw tree-ring width series but this is not optimal to remove ontogenetic effect. A mean function will not change the shape of the series, therefore the declining growth trend will still be present after detrending…which makes detrending useless in this case. A modified negative exponential would be better, especially if the authors decide to only keep the decreasing part of the raw ring width series (cf. L148). Morevoer, I think that the mean detrending is mostly responsible for the fact P* is higher than RWI over the most recent period and lower in the earliest years (L242-243).

Why no power transformation has been performed before detrending TRW to stabilize the variance? Why no autocorrelation removal has been performed after detrending?

These two steps (not perforemd here) are usually performed to maximize the link between growth patterns and climate. The authors say themselves that P* was designed to reflect climate impacts on radial growth (L211). It would therefore be more optimal to compare it to properly standardized RWI (incl. power transformation, negative exponential detrending and autocorrelation removal).

- Computation of the $Hm_{(ICP)}$

Please provide a map of the location of the sites where the paired D and H measurements were taken and from which $Hm_{(ICP)}$ computation are derived. Are these sites representative, in term of stand characteristics and environmental gradient, of your study sites (those extracted from the ITRDB)? For me estimation of asymptotic maximum height should not be performed using the overall dataset, but should be performed along a climatic gradient or at least along an latitudinal gradient.

SPECIFIC COMMENTS

- Methodology

I have had hard time to understand at which stage the analyses move from a tree-level to a site-level. Some of the described analyses are hard to follow also because the authors seldom describe the statistical tools they used. One example of this lack of clarity is the 3.3. section. We don't know how the contribution of each tree to the site P* was computed.

L164- Which resolution does this dataset have?
Why using E-OBS when the WFDEI data look much more resolved temporally and spatially?

L195. Are you talking about standardized or raw annual diameter increments here?

- Statisitcal tools and methodology

Similarly I find that almost none of the methods presented described the statistical environment (program, packages, functions) in wihch analyses were performed.

Section 3.3. please provide the packages or statistical environment used for these analyses

L205. Please provide the statistical tools you used to standardize the raw tree-ring width series.

L262. The notion of first maximum has not be defined yet. Please define it or rather the 'ontogentic turning point' or 'peak in radial growth' you mention.

-References

I think that the authors are not enough supporting what they write by scientific references:
L101-102 Assumption 1: environmental effects presumed to influence the growth of all trees. Do you have a reference for this?

L51. Use a reference to support that tree growth is sensitive to environmental factors (even if we all know it).

L58-59 please use a reference for this.

L64. Please use a reference for this.

L68 Please use a reference for this

L77. Please use a reference for this

L171. Please use a reference for this statement. Who has defined the 'effective carbon accumulation year'?

- Others comments

L148. Why would you remove earlier rings (youngest trees) if your main target is to correct for the sampling biais over old trees?

L 50. "because of their chronological accuracy"> please mention that this is achieved by applying the principles of crossdating.

L82. 'we require a reliable method to process the large volume of tree-ring data'. I don't agree with this statement. Numerous methods already exist for big data analyses (ANN, kNN, RF etc.) and these are already quite reliable.

L87-88. This seems logical to me. When you go sampling, no matter which tree you sample, the trees are older over 1970-2000 than over 1940-1969…Regardless of the structure of the stands…

L97-98. This statement can be argued. There is an endless discussion regarding the age vs size effect on growth trends. The discussion is not over. Please see XXX.
However, no one would argue with the fact that size is easier to measure and more accurately measured than age at a certain tree height.

L278-279. Conventional approaches are also based on equal contribution/weight of all tree to the master chronology…

L284. Add 'positive response to alpha'

L290. Please compare the model $P^*= f(mGDD_5)$ with the model $P^*=f(mGDD_5 + mGDD_5^2)$, to prove that applying a parabolic regression is better than a simple linear regression.

Figure 3.
a- Larger differences between RWI and $P^*$ are observed at more productive sites. Do you have any explanation for this?
b- Please mention in your caption the change in x and y-axis scale across panels.

Figure 6.
a- I identify subgroups in the points clouds of some panels (ex. upper left panel on the extreme right of the cloud). Is this link to geography? Could you plot the sites from North America and Eurasia in different colors? It would be very interesting to provide us with the same analyses by biome (mediterranean, temperate, boreal). See my major comment about this.
b- I notice a higher variability in $P^*$ in the middle of the gradient in the middle panels of PPFD5. Do you have any explanation for this?

Figure 7.
a- I am surprised to see that the p value of the regression in panels $CO^2$ vs Pinus 1970-2000 and $CO^2$ vs Picea1970-2000 is significant. To me it seems that the linear regression does not present any slope… Do you have any explanation for this?

---

## Referee Comment (RC2) · Anonymous Referee #2 · 13 May 2019

In this study, the authors present a new approach to remove potential ontogenetic growth trends and demographic biases from tree-ring data. Such trends can confound the detection of trends caused by environmental change. The aim of the authors is highly relevant, the datasets used are large, figures are of high quality, and the manuscript is well written.

However, my main critique at this point is that the rationale and logic for choosing this particular model remains poorly explained. More importantly, why this new approach would account for ontogenetic growth changes and potential demographic biases is not discussed. I struggled to understand by what mechanism their new approach would account for ontogenetic trends and biases. Lastly, the authors claim in L247-248 and L305-307, that "P* has effectively reduced the effect of sampling biases as well as accounting for ontogenity", but how can this be proven? Does this not require that these effects were known and quantified prior? The authors use empirical data, but I expect

this can only be really tested with simulated data that includes a known ontogenetic and demographic effect.

I also have some minor comment that I hope will help to improve the manuscript.

L39. Suggest ending the abstract with a concluding sentence(s).

L53-54. This pattern may be present in light-demanding trees, recruiting in high light conditions. For shade tolerant trees, opposite patterns can be observed: initial suppressed growth, with higher growth rates later in life.

L58-74. I totally agree that it is in general difficult to separate growth trends caused by ontogeny from those caused by global change. And it is true that RCS may also remove changes caused by the latter. However, van der sleen et al. (2015) controlled for ontogenetic effects by sampling trees at a fixed diameter (so no detrending). Nonetheless, regardless of how ontogenetic changes are accounted for, tree-ring data can also be affected by various sampling biases which, in addition to ontogenetic and environmental change, can cause trends in the data. I think this should be pointed out more clearly here. For example, here one wonders if the focus of the paper will be on accounting for ontogenetic effects and/or for sampling biases? This should not be mixed.

L271. I think the figure reference is wrong here (there are no left panels in Fig. 5).

L307. Here and elsewhere in the manuscript, I think the model used is relatively complex (and requires quite a lot of parameters) and is thus not so "straightforward" as claimed.

---

## Author Comment (AC1) · 30 Aug 2019

Dear Editor,

We were very grateful to receive such thorough reviews for our manuscript. We appreciate both the two reviewers' critical and constructive comments and suggestions, which we believe would be very helpful in lifting both the quality and readability of our manuscript.

Sincerely,
Guangqi Li on behalf of all of the co-authors

The response to reviewer's comments are using the normal font, with reviewers' comments italic.

Referee #2
*In this study, the authors present a new approach to remove potential ontogenetic growth trends and demographic biases from tree-ring data. Such trends can con- found the detection of trends caused by environmental change. The aim of the au- thors is highly relevant, the datasets used are large, figures are of high quality, and the manuscript is well written.*
*However, my main critique at this point is that the rationale and logic for choosing this particular model remains poorly explained. More importantly, why this new approach would account for ontogenetic growth changes and potential demographic biases is not discussed. I struggled to understand by what mechanism their new approach would account for ontogenetic trends and biases. Lastly, the authors claim in L247-248 and L305-307, that "P\* has effectively reduced the effect of sampling biases as well as accounting for ontogenity", but how can this be proven? Does this not require that these effects were known and quantified prior? The authors use empirical data, but I expect this can only be really tested with simulated data that includes a known ontogenetic and demographic effect.*

This is a really helpful suggestion. We agree that using simulated data, with known ontogenetic and demographic structure could be the better way to show Pstar could account for ontogenetic growth changes and potential demographic biases, which might be a big problem when using traditional dendro detrending method, especially in the respect of reserving long-term climate signals. Because the long-term trended climate signals might be removed along with the detrending process, which is caused by the widely distributed "demographic biases" in tree ring sampling. This analysis will be a very important session in the revised ms.
Furthermore, we will add more details about the method for Pstar extraction, especially in session 3.3 Estimation of P\*. The R code for this process will also be submitted as the supplementary information in the revised ms. There would be detailed instructions on how that 2-step regression was done for the estimation about long-term mean stand-leval and individual-level P\*, and the time-series of stand-level P\*.
The next is about using simulated ring width to prove "P\* has effectively reduced the effect of sampling biases as well as accounting for ontogeny".
In this analysis, the climate was set as a long-term drying trend from soil moisture (-0.1%/year), adding with random noise to make the climate more realistic. Simulated ring width is from T model (Li et al. 2014). The simulated ring width and its ontogenetic effect (growth) are shown in the following figure, with the comparison of the reduced GPP, which is caused by the long-term trended soil drought.

[Figure]

There are 40 trees simulated by T model, with 8 age-cohorts (5 trees/recruitments for each age cohort). The recruitment was set majorly happened at the beginning period, which is to mimic the mentioned sampling demographic bias, which is older/bigger trees are more likely to be sampled. Here is the simulated ring width for all 40 trees.

[Figure]

If we use the Reginal Growth Curve (RCS) method to detrend those ring width, the following growth curve was applied. It is obvious that the estimated ontogenetic growth curve by RCS is much steeper than the "real" ontogenetic growth curve. This is caused by the long-term climate, and this signal could be partially/fully removed by this RCS detrending.

[Figure]

As expected, the final RCS chronology doesn't reserve the long-term drought signal at all.

[Figure]

Then, we applied the more traditional dendrochronology method – Negative Exponential Curve or linear detrending, and included log transformation and removal autocorrelation when building the final chronology. As either the exponential line or the straight line was chose to represent the ontogenetic growth curve for each tree. The long-term drought signal has already been included in this "ageing curve".

[Figure]

Here is the final chronology using NE method. Both the power transformed, and removal autocorrelation chronologies were built to check whether the long-term trend is reserved. And neither of the NE chronology is working. And because either the exponential line or linear line is not that flexible as the RCS method, it underestimates the growth both at the beginning and the ending period. Thus, it turns out that the estimated soil drought is weaker at the two ends, and stronger drought for the middle part.

[Figure]

Then Pstar was applied to extract this long-term trend climate/drought signal. Compared with the two former detrending methods, Pstar is the only one that could save this decreasing soil moisture trend. Although it fails at the earlier period. It is majorly caused by the different simulation/assumption in T model and Pstar, which Pstar requires trees are big enough, and T model can simulate the whole life stage of the tree. This also relates to the peak growth position in P* estimation. A more conservative estimation about the peak could improve P*'s performance, but at the expense of losing record length.

Just to clarify one point about Pstar extraction, there are no specific steps of building an ontogenetic growth curve and then remove it from raw ring width data. The way of Pstar estimation is to use the theoretic-based regression, which is majorly about functional geometry constraint carbon allocation, to estimate how much GPP is needed for a specific stem growth at a certain size of a tree.

[Figure]

*I also have some minor comment that I hope will help to improve the manuscript.*
*L39. Suggest ending the abstract with a concluding sentence(s).*
Thanks for the suggestion. We will add the concluding sentence in the revised abstract.

*L53-54. This pattern may be present in light-demanding trees, recruiting in high light conditions. For shade tolerant trees, opposite patterns can be observed: initial suppressed growth, with higher growth rates later in life.*
We agree. And this gap releasing growth pattern usually have at least two growth peaks. And usually, the higher peak is after the gap releasing. Then the growth rate will gradually slow down as tree grows bigger. And this second phase of growth pattern is similar to the normal light-demanding trees in high light conditions. This means we could still use the second stage of growth data for Pstar estimation. We will add this gap releasing growth pattern in the new introduction.

*L58-74. I totally agree that it is in general difficult to separate growth trends caused by ontogeny from those caused by global change. And it is true that RCS may also remove changes caused by the latter. However, van der sleen et al. (2015) controlled for onto- genetic effects by sampling trees at a fixed diameter (so no detrending). Nonetheless, regardless of how ontogenetic changes are accounted for, tree-ring data can also be affected by various sampling biases which, in addition to ontogenetic and environmental change, can cause trends in the data. I think this should be pointed out more clearly here. For example, here one wonders if the focus of the paper will be on accounting for ontogenetic effects and/or for sampling biases? This should not be mixed.*
Thanks a lot for the suggestions. We will clarify that our method is to account to deal with the wrongly removing long-term trend caused by the sampling biases.
As mentioned above, Pstar is developed to deal with the issue that traditional detrending methods have high possibility to wrongly remove long-term climate trends, which is majorly caused by the sampling biases, the demographic bias. We will clarify and strengthen this point in both the introduction and the conclusion part.

*L271. I think the figure reference is wrong here (there are no left panels in Fig. 5).*
Apologies for the typo, and this should be Fig. 4

*L307. Here and elsewhere in the manuscript, I think the model used is relatively complex (and requires quite a lot of parameters) and is thus not so "straightforward" as claimed.*
We will improve this, by adding more detais in the method session (3.3 P* estimation). Meanwhile, we will also upload the R code for P star extraction with the revised manuscript. To clarify, there are only two parameters needed for Pstar extraction (if the peak growth position estimation is using the simple peak of ring width), which is *a* (the initial slope of height to diameter) and Hm (asymptotic maximum height). We have already analysed the value of *a* and Hm impact on the final Pstar (figure 4). It showed that Pstar would be heavily infuluenced by the accuracy of *a* and Hm.
To estimate Pstar, a function from limited lines of R code (dozens of lines) could be very easily done, with the only input of *a* and Hm.

**Referee1**
GENERAL COMMENTS
*This well written paper falls in the scope of the journal. I found pleasant to discover a new born methodology to handle tree-ring series in order to perform unbiased climate-growth analyses. However, even if this novel method looks appealing from a first perspective, it requires quite a substantial amount of parameters to be applied, compared to conventional methods (L307). I am not convinced that the application of this novel method is straightforward, despite what the authors say (cf. L307), particularly for other less common tree species or genii.*
As mentioned above, we will add more details in the method session (3.3 P* estimation). Meanwhile, we will also upload the R code for P star extraction with the revised manuscript. To clarify, there are only two parameters needed for Pstar extraction (if the peak growth position estimation is using the simple peak of ring width), which is *a* (the initial slope of height to diameter) and Hm (asymptotic maximum height). And the final P* is quite resilient to the value of *a* and Hm.
The R code attached with the revised ms will show how straightforward Pstar estimation could be. This runs more or less the same as the conventional detrending methods.

*The high positive correlation between P* and RWI makes me think again that the classical detrending applied in dendrochronology would still be easier and simpler to use. To really advocate in favour of the use of the herein presented method, it would require to compare the climate-growth analyses presented in Figs 6 and 7 with similar climate-growth analyses computed with 'traditionally detrended RWI.*

We agree that this will be very helpful to prove Pstar's ability in reserving the long-term climate trend under current sampling biases. We will add this analysis in the revised ms, as the second evidence to support the use of Pstar.

Here are the climate-growth analysis using the traditionally detrended RWI, which are RCS, power-transformed Negative Exponential (NE), and the removal autocorrelation one.
- The parabola for $mGDD_5$ for all of the traditional detrending methods is squeezed. And the vertex turns to much less stable than Pstar, with larger uncertainty (SD).
- Meanwhile, the variance explained by the total model ($R^2$) was reduced when using any of the traditional methods. $R^2$ in order is: Pstar (0.8-0.9) >> RCS (0.4-0.5) > NE-removal_autocorrelation (0.3-0.5) > NE-power_transformed (0.23-0.5). This could be another strong evidence to prove Pstar is conserver much "better" climate signals in the final chronologies.
- The next 3 groups of figures are the climate-growth analysis done in Figure 6, but using chronologies generated by RCS, NE-power transformed, and removal autocorrelation.

[Figure]

*I have some major concerns that I hereafter discuss:*

*- Study area and environmental gradient*

*I am skeptical regarding the gradient over which they perform their analysis. The authors take two conifer tree genus from the northern hemisphere as examples, Pinus and Picea species. The tree populations include sites and pine and spruce species from the mediterranean, temperate and boreal contexts that respond quite differently to climate (e.g. Pinus laricio spp. vs Pinus sylvestris). For example, it is well known that growth responses to summer temperature move from being negative in warm and dry contexts (mediterranean biome) to positive in cold and wet environment (boreal biome). So I would like to see the analyses, especially the climate-growth analyses, ran for each genus over the three above-mentionned biomes.*

*This would be even more needed as the authors assume that "bioclimatic parameters have a consistent impact on growth" (cf. L212-213). This assumption is wrong, and it is even more wrong to assume that this is true regardless of sampling period or targeted tree species (cf. L213). Growth responses to climate are by nature not constant through time and space for a given tree species, due to non-linear interactions between environmental factors (e.g. climate and soil). The authors mention themselves (cf. L 220-221) that mGDD5 has a non-linear impact on P\*.*

We thank the reviewer pointing this out, which is also what we want to demonstrate in our ms. And we believe our original idea is the same as what was stated by the reviewer, which is about the changing responses to a specific climate variable for different types of climate/environmental conditions. This is also the major reason/theory that we followed to design the climate-growth analysis via multiple GLM models. In plant physiology theory, tree growth is jointly controlled by multiple climates. One specific climate variable's impact on tree growth is depended on the whole climate conditions. So, in such a way, we share the same idea about a single climate variable's impact on tree growth. And we are happy and have done the biome based climate-growth analysis, which is suggested by the reviewer. We will revise our ms to clarify this point.

Here are the climate-growth analysis for both the two genera over the three biomes.

- Because of site availability, in Mediterranean climate region, there is only *Pinus*. Temperate *Pinus* are majorly at North America. "Boreal" *Pinus* was set in the Scandinavia, although it is really not boreal. Thus, the three biomes for *Pinus* are defined as: 1) Mediterranean (lat: 25 - 45 and lon: -10 - 40), 2) temperate in North America (lat: 35 - 55 and lon: -140 - -30), 3) boreal in Scandinavia (lat: 60 - 75; lon:0 - 40)
- There is no Mediterranean *Picea*, at least in our dataset. We divid data into a) temperate European *Picea* region (lat: 25 - 62 and lon: -10 - 40), b) temperate America *Picea* region (lat: 35 - 57 and lon: -140 - -70); and c) boreal *Picea* in north America (lat: 57 - 70; lon: -180 - -50)

[Figure]

- It is obvious that in the Mediterranean region the growth response of *Pinus* to temperature is majorly negative. Meanwhile, soil moisture's impact is very strong.

[Figure]

- For temperate *Pinus* in North America, growth response to mGDD5 is consistent among 3 periods. and its impact on *Pinus* growth is changing from flat to negative.

[Figure]

- Those "boreal" *Pinus* in Scandinavia is not really at the "boreal" climate, because the warm and humid maritime climate in northern Europe.
- It is obvious that solar radiation (PPFD5) is much lower than the other regions. The positive slopes of PPFD5 are significantly positive. In contrast, soil moisture is not a limiting factor.
- Response to mGDD5 is almost flat, falls at the near vertex region.

[Figure]

Europe Picea

- For the Europe temperate *Picea* sites, response to mGDD5 is flat, indicates it falls at the vertex of parabola.

[Figure]

America Temperate Picea

- A more negative than flat slope of mGDD5 was shown for temperate *Picea* in North America.

[Figure]

America Boreal Picea

- Boreal Picea, shows a clear upward slope of mGDD5.

Summary of the climate-growth analysis for 3 different biomes:

- Temperature's impacts on tree growth are roughly showing what the reviewer's assumption, which is changing from negative in the south warm Mediterranean, to flat-to-negative in the temperate region, and to positive to the boreal forest.

- However, because of the limited climatic range, the controls (the value of the slope) from those climate variables are showing a uniform pattern over different biomes. In such case (smaller climate range), the unimodal regression might not be as good as the linear regression. But it still shows the roughly expected large-scale patterns.
- There might be another problem with this biome analysis, which is about the way we define biomes. Here we simply used "regional" coordinates (lat and lon) to categorise biomes. This is not true in the real world, especially for those mountain sites. For example, not all of the sites around the Mediterranean is Mediterranean vegetation type. The tree line (upper limit) in the Mediterranean region should not belong to Mediterranean biome, and it might fall to the positive-mGDD5-response climate region.

We found a compromised way might also be able to support our shared idea about the changing response to a specific climate variable. It is not directly analysing data within those biomes. Let's look at the distribution of climate space of our sites. As we already found that 9-11 oC is around the vertex of mGDD5's impact on tree growth. So we could divide the sites into 3 parts by 10 oC and 13 oC (below figure). 10 oC is the middle of 9 oC and 11 oC. 13 oC is just to separate boreal sites from temperate region. If those 3 groups, which is showing positive, flat, and negative to mGDD5 respectively, could roughly cover the 3 biomes, that could also prove our shared changing impact on tree growth.

- The blue sites, which should be the positive-response region, falls majorly at the boreal and high mountain area. And these region's soil moisture is usually high.
- Red sites – negative-response region – are not only at the Mediterranean. But they are majorly distributed in the southern part and with relatively lower soil moisture.
- Orange sites fall at the top of the parabola, which means mGDD5's impact is flat/not important. For those regions, soil moisture is still high enough. And these are normally the temperate regions.

[Figure]

*- 'the sampling biais of age distribution'*
*The authors ground their study in correcting the under-representation of young trees in growth chronologies. Therefore I have hard times to understand why the authors remove all rings before the turning point in growth trends (i.e. all juvenile rings). By doing so, they also contribute to the sampling biais of age distribution in their site chronologies. To me it seems that this removal of juvenile growth rings is only done to allow the approximation done in Appendix C. In traditional detrending, juvenile growth rings are kept.*

As we clarify above about Pstar extraction, there is no such a step of building an ontogenetic growth curve and then remove it from raw ring width data. The way of Pstar estimation is to use the theoretic-based regression, which is majorly about functional geometry constraint carbon allocation, to estimate how much GPP is needed for a specific stem growth at a certain size of a tree. Removing juvenile growth is not to correct demographic bias. It is the requirement of Pstar estimation, which requires trees are not too small.

*A question arises: What is the average number of years (juvenile rings) removed by this step? How does age distribution look like after this step compare to the original age distribution (Fig. 1b)?*

The average number of years (juvenile rings) removed by this step is a bit lower than 50 years for both of the two genera. This is the summary based on the peak was identified by using the "theoretic peak" method, which is more conservative, with a smaller variation. And if we use the maximum value to identify the peak (simple peak), the mean value is around 25 years for both genera, but with a much bigger variation.

*Pinus*:                                                            *Picea*:

[Figure]

Again, this peak identification followed with the removal juvenal tree growth data, won't change the distribution of the age/size. To clarify, Pstar extraction should be for "not too small" trees. This is why we need to identify the peak growth position. We agree that the traditional detrending method might be able to use those juvenile rings.

However, the major problem, we concerned about those methods, is that it might have already removed the long-term trended climate signals when detrending is applied. In the meantime, as mentioned in the experiment using simulated ring width for the detrending comparison, the traditional methods estimation for juvenile growth are not that good.

*- Simplification of equations*
*The authors are simplifying quite substantially the equations they present in Appendix C based on different assumptions. Would these assumptions be applicable to other tree species, e.g. broadleaves species? If not, the herein presented method couldn't be applied to some tree species, and therefore would be useless in an more isolated geographical context.*

Those simplifications are majorly based on mathematics, rather than on plant physiology. The original model (T model, Li et al., 2014) has been applied for multiple species over quite different climate conditions around the world, including *Pinus* in China, *Juniperus* in North America, *Callitris* in Australia, *Cedrus* in North Africa. This model was built based on plant physiology and well-known function geometry. Theoretically, all C3 tree species could be applied for this model.

*- Detrending of tree-ring series*
*The authors use a mean fonction to detrend raw tree-ring width series but this is not optimal to remove ontogenetic effect. A mean function will not change the shape of the series, therefore the declining growth trend will still be present after detrending...which makes detrending useless in this case. A modified negative exponential would be better, especially if the authors decide to only keep the decreasing part of the raw ring width series (cf. L148). Morevoer, I think that the mean detrending is mostly responsible for the fact P\* is higher than RWI over the most recent period and lower in the earliest years (L242-243).*
*Why no power transformation has been performed before detrending TRW to stabilize the variance? Why no autocorrelation removal has been performed after detrending?*
*These two steps (not perforemd here) are usually performed to maximize the link between growth patterns and climate. The authors say themselves that P\* was designed to reflect climate impacts on radial growth (L211). It would therefore be more optimal to compare it to properly standardized RWI (incl. power transformation, negative exponential detrending and autocorrelation removal).*

Thanks for the suggestion. This has been done and the results are shown above. This comparison will be added to the revised ms.

*- Computation of the Hm(ICP)*

*Please provide a map of the location of the sites where the paired D and H measurements were taken and from which Hm(ICP) computation are derived. Are these sites representative, in term of stand characteristics and environmental gradient, of your study sites (those extracted from the ITRDB)? For me estimation of asymptotic maximum height should not be performed using the overall dataset, but should be performed along a climatic gradient or at least along an latitudinal gradient.*

We agree the best way estimating Hm (asymptotic maximum height) would be along the climatic gradient or even for a single site. We estimate Hm and *a* using the European forest inventory data. We have tried to find whether there is a clear and simple patter for *a* (and Hm) with climatic gradients. *a* and Hm was estimated for each site (figure below). Unfortunately, we didn't found a clear pattern of both *a* and Hm over Europe. To test whether the choice of *a* and Hm would have a big impact on Pstar simulation, sensitivity analysis using plus and minus 50% of Hm and *a* has been done. And it showed that their impact on the final P* is very small (Figure 4). Considering data availability and its relatively small influence on final Pstar, we adopt the method. In the meantime, the idea of climate controls on Hm is also applied for those lower sites, which is constraint by using GPP as a predictor.

[Figure]

[Figure]

**SPECIFIC COMMENTS**

*- Methodology*

*I have had hard time to understand at which stage the analyses move from a tree-level to a site- level. Some of the described analyses are hard to follow also because the authors seldom describe the statistical tools they used. One example of this lack of clarity is the 3.3. section. We don't know how the contribution of each tree to the site P\* was computed.*

As mentioned, we will add more details about the method for Pstar extraction, especially in session 3.3 Estimation of P\*. The R code for this process will also be submitted as the supplementary information in the revised ms. There would be detailed instructions on how that 2-step regression was done for the estimation about long-term mean stand-level and individual-level P\*, and the time-series of stand-level P\*.

*L164- Which resolution does this dataset have? Why using E-OBS when the WFDEI data look much more resolved temporally and spatially?*

Both CRU and WFDEI are using $0.5^{\circ}$ x $0.5^{\circ}$ resolution. WFDEI data are 3-hourly. To calculate mean temperature during growth season, and PPFD, it performs much better than the monthly data of CRU. We only used CRU monthly climate data to estimate the annual soil moisture status. Because the monthly resolution for estimation the annual soil moisture is sufficient. We will make this clearer in the revised ms. We didn't use E-OBS climate data.

*L195. Are you talking about standardized or raw annual diameter increments here?*
This is the raw ring width. We will clarify this in the revised ms.

*- Statisitcal tools and methodology*
*Similarly I find that almost none of the methods presented described the statistical environment (program, packages, functions) in wihch analyses were performed.*
We are using R, and the major package of dplR for traditional detrending, and lmer for mixed model regression. These information will be added in the revised version.

*Section 3.3. please provide the packages or statistical environment used for these analyses L205. Please provide the statistical tools you used to standardize the raw tree-ring width series.*
We applied the R package "dplR" for those traditional detrending and chronology building. We will add those information with the revised ms.

*L262. The notion of first maximum has not be defined yet. Please define it or rather the 'ontogentic turning point' or 'peak in radial growth' you mention.*
We will define it here in the revised ms.

*-References*
*I think that the authors are not enough supporting what they write by scientific references: L101-102 Assumption 1: environmental effects presumed to influence the growth of all trees. Do you have a reference for this?*
Sorry for the confusing. Here we mean the fixed effect is the mean stand-level environmental conditions. We will clarify this in the revised ms.

*L51. Use a reference to support that tree growth is sensitive to environmental factors (even if we all know it).*
*L58-59 please use a reference for this. L64. Please use a reference for this. L68 Please use a reference for this*
*L77. Please use a reference for this*
*L171. Please use a reference for this statement. Who has defined the 'effective carbon accumulation year'?*
We will add the reference for those lines in the revised ms.

*- Others comments*
*L148. Why would you remove earlier rings (youngest trees) if your main target is to correct for the sampling biais over old trees?*
As mentioned above, the peak identification followed with the removal of juvenal tree growth data, won't change the distribution of the age/size. To clarify, Pstar extraction should be for "not too small" trees. This is the only reason why we need to identify the peak growth position. It is using theoretic-based regression, rather than using the existing data to estimate growth trend, then detrending, to get the climate signals from tree ring data. We will clarify this point in the revised ms.

*L 50. "because of their chronological accuracy"> please mention that this is achieved by applying the principles of crossdating.*
We will add this information in the revised ms.

*L82. 'we require a reliable method to process the large volume of tree-ring data'. I don't agree with this statement. Numerous methods already exist for big data analyses (ANN, kNN, RF etc.) and these are already quite reliable.*
We will revise this sentence by emphasising "researving long-term trended climate signal".

*L87-88. This seems logical to me. When you go sampling, no matter which tree you sample, the trees are older over 1970-2000 than over 1940-1969...Regardless of the structure of the stands...*
This is the also the point of sampling bias. If sufficient young trees could be sampled for more recent period, those age distribution pattern will be changed more evenly. We will clarify this in the revised ms.

*L97-98. This statement can be argued. There is an endless discussion regarding the age vs size effect on growth trends. The discussion is not over. Please see XXX.*

*However, no one would argue with the fact that size is easier to measure and more accurately measured than age at a certain tree height.*

*L278-279. Conventional approaches are also based on equal contribution/weight of all tree to the master chronology…*

*L284. Add 'positive response to alpha'*

Thanks for the reviewer, we will be more careful and rephase these point in the revised ms.

*L290. Please compare the model P\*= f(mGDD5) with the model P\*=f(mGDD5 + mGDD5$^2$), to prove that applying a parabolic regression is better than a simple linear regression.*

Thanks for the suggestion. As suggested, the two regression are done for the comparison. The parabolic regression performs better in both the higher fixed effect variance explanation (R2-fixed), and the total R2 of the whole model. For instance, the marginal coefficient of determination for the model *P\*=f(mGDD5 + mGDD52)* for the whole period of 1940-2000 is 0.0734, while that for the model *P\*= f(mGDD5)* is 0.0679; and the conditional coefficient of determination for the model *P\*=f(mGDD5 + mGDD52)* for the whole period of 1940-2000 is 0.850, while that for the model *P\*= f(mGDD5)* is 0.845.

*Figure 3.*

   *a- Larger differences between RWI and P\* are observed at more productive sites. Do you have any explanation for this?*

This should not be the case. The difference between RWI and P\* is decided by how those sampling's age distribution and whether there is a obvious long-term climate influencing tree growth.

   *b- Please mention in your caption the change in x and y-axis scale across panels.*

We will add this information in to the caption

*Figure 6.*

*a- I identify subgroups in the points clouds of some panels (ex. upper left panel on the extreme right of the cloud). Is this link to geography? Could you plot the sites from North America and Eurasia in different colours? It would be very interesting to provide us with the same analyses by biome (Mediterranean, temperate, boreal). See my major comment about this.*

Thanks for the suggestion. As mentioned above, the biomes climate-growth analysis has been done. There results will be included in the revised ms.

*b- I notice a higher variability in P\* in the middle of the gradient in the middle panels of PPFD5. Do you have any explanation for this?*

This middle ranged PPFD5 are more likely fell into the temperate forest region, which is both the major distribution for *Pinus* and *Picea*. This residual plot is showing under certain fixed other climate conditions, what's the impact of PPFD5 on tree growth. This bigger range should be majorly caused by the major distribution for both the two genera, and it should also mean there are a bigger range for mGDD5 and soil moisture in these regions.

*Figure7.*

*a- I am surprised to see that the p value of the regression in panels CO$_2$ vs Pinus 1970-2000 and CO$_2$ vs Picea1970-2000 is significant. To me it seems that the linear regression does not present any slope… Do you have any explanation for this?*

Based on my understanding, the *P-value*, which shows the significance of the slope, is not necessarily to be linked with the degree of slope. Although the slope is significant, CO2's impact on tree growth is still small, based on our analysis.